# Single-molecule level control of host-guest interactions in metallocycle-C$_{60}$ complexes

Jian-Hong Tang [1,2,9], Yueqi Li [3,9], Qingqing Wu [4,9], Zixiao Wang [5,9], Songjun Hou[4], Kun Tang [1,6], Yue Sun[2], Hui Wang[5], Heng Wang [7], Cheng Lu[8], Xiang Wang[8], Xiaopeng Li [7], Dong Wang[8], Jiannian Yao[1], Colin J. Lambert [4*], Nongjian Tao[3,5*], Yu-Wu Zhong [1,6*] & Peter J. Stang[2*]

Host—guest interactions are of central importance in many biological and chemical processes. However, the investigation of the formation and decomplexation of host—guest systems at the single-molecule level has been a challenging task. Here we show that the single-molecule conductance of organoplatinum(II) metallocycle hosts can be enhanced by an order of magnitude by the incorporation of a C$_{60}$ guest molecule. Mechanically stretching the metallocycle-C$_{60}$ junction with a scanning tunneling microscopy break junction technique causes the release of the C$_{60}$ guest from the metallocycle, and consequently the conductance switches back to the free-host level. Metallocycle hosts with different shapes and cavity sizes show different degrees of flexibility to accommodate the C$_{60}$ guest in response to mechanical stretching. DFT calculations provide further insights into the electronic structures and charge transport properties of the molecular junctions based on metallocycles and the metallocycle-C$_{60}$ complexes.

---

[1] Beijing National Laboratory for Molecular Sciences, Key Laboratory of Photochemistry, CAS Research/Education Center for Excellence in Molecular Sciences, Institute of Chemistry, Chinese Academy of Sciences, Beijing 100190, China. [2] Department of Chemistry, University of Utah, 315 South 1400 East, Room 2020, Salt Lake City, UT 84112, USA. [3] Biodesign Center for Bioelectronics and Biosensors, Arizona State University, Tempe, Arizona 85287, USA. [4] Department of Physics, Lancaster University, Lancaster LA1 4YB, UK. [5] State Key Laboratory of Analytical Chemistry for Life Science, School of Chemistry and Chemical Engineering, Nanjing University, Nanjing 210023, China. [6] School of Chemical Sciences, University of Chinese Academy of Sciences, Beijing 100049, China. [7] Department of Chemistry, University of South Florida, 4202 East Fowler Avenue, Tampa, FL 33620, USA. [8] CAS Key Laboratory of Molecular Nanostructure and Nanotechnology, Institute of Chemistry, Chinese Academy of Sciences, Beijing 100190, China. [9] These authors contributed equally: Jian-Hong Tang, Yueqi Li, Qingqing Wu, Zixiao Wang. *email: c.lambert@lancaster.ac.uk; njtao@asu.edu; zhongyuwu@iccas.ac.cn; stang@chem.utah.edu

The discovery of crown ether and cryptand in 1967 as macrocyclic ligands for alkali metal cations laid the foundation of modern host−guest and supramolecular chemistry[1–3], which aims to understand various noncovalent intermolecular interactions in chemical and biological entities[4–8]. Host–guest interactions have now been exploited in a wide range of disciplines, including chemosensing[9,10], catalysis[11], drug delivery[12], nanosciences[13,14], and molecular devices[15,16]. These previous works focused on host–guest ensemble behaviors of bulk materials. Studying how host–guest complexes behave at the single-molecule level opens new prospects to understand the fundamentals of supramolecular chemistry by providing critical information about the strength and flexibility of host–guest interactions.

The scanning tunneling microscopy break junction (STMBJ) technique allows measurement of charge transport properties of single-molecules involving covalent bonds[17–27]. The technique has been recently extended to supramolecular systems including charge−transfer pairs[28], π-stacked dimers[29], rotaxanes[30,31], or host–guest interactions[32]. Mechanically controllable break junction (MCBJ) or STMBJ experiments are well-developed to reveal the electrical and mechanical properties of the molecular structure and junction conductances[33–38].

Herein, we measure the conductance and study the charge transport in three supramolecular metallocycles and their host–guest complexes with $C_{60}$ with STMBJ. The coordination-driven self-assembly yields different metallocycles with precisely controlled shape and cavity size, which allows us to modulate the strength of the host–guest interaction. The conductance of the metallocycles with a suitable cavity size is enhanced by an order of magnitude when complexed with the $C_{60}$ guest. The molecular conductance properties of the metallocycles and their host–guest complexes are further rationalized by theoretical calculations. The direct monitoring of the size-dependent decomplexation was obtained by mechanically stretching the host-guest complexes at the single-molecule level, which switches the conductance back to the free-host level. This demonstrates the capability of host-guest supramolecular systems to modulate molecular conductance.

## Results and discussion

**Molecular design, synthesis, and characterization.** Three organoplatinum(II) metallocycles, **5**, **6**, and **7** with different shape and cavity size were designed and synthesized for the host–guest interaction studies and conductance measurements (Fig. 1), via the coordination-driven self-assembly of carefully selected Lewis acid metallic acceptors and Lewis base organic donors with the formation of multiple platinum–pyridine nitrogen bonds[33,34,39–43]. The [2 + 2] assembly of the 120° donor **1** with the 60° diplatinum acceptor **2** or **3** gave the molecular rhomboid **5** and **6**, respectively. The hexagonal metallocycle **7** was obtained from the [3 + 3] assembly of **1** with the 120° diplatinum acceptor **4**. These products were obtained in nearly quantitative yields (see details in the Supplementary Methods). The uncoordinated pyridine groups present in these metallocycles are used to contact the gold metal electrodes to form single-molecule junctions.

The formation of the above metallocycles were established by $^1H$ NMR, $^{31}P\{^1H\}$ NMR, $^1H−^1H$ COSY NMR spectroscopy, and electrospray ionization time-of-flight mass spectrometry (ESI-TOF-MS) (Supplementary Figs. 1–12). In the $^1H$ NMR spectra, distinct downfield shifts of the $\alpha$ and $\beta$ protons of the two terminal pyridine groups of **1** were observed when it was incorporated into these metallocycles (Supplementary Figs. 1, 5, and 9). The $^{31}P\{^1H\}$ NMR spectra show sharp singlet signals at $\delta$ of 14.54, 16.81, and 17.03 ppm for **5**, **6**, and **7**, respectively, with concomitant $^{195}Pt$ satellites, indicating their single-phosphorus

environments (Supplementary Figs. 2, 6, 10). ESI-TOF-MS data provided further evidence on the formation of these assemblies (Fig. 1; Supplementary Figs. 4, 8, and 12). The peaks at m/z of 897.61, 929.63, and 989.57 Da, corresponding to the positively charged metallocycle frameworks, $[5 − 3OTf]^{3+}$, $[6 − 3OTf]^{3+}$, and $[7 − 4OTf]^{4+}$, respectively, were isotopically resolved to agree with their theoretical distributions (Fig. 1).

The density functional theory (DFT)-optimized structures of **5** and **6** show a rhomboid geometry with a cavity size about 10.87 and 14.10 Å, respectively (Supplementary Fig. 13). In contrast, the optimized structure of **7** exhibits a hexagonal configuration with a much larger cavity size of 22.85 Å. Since the van der Waals diameter of $C_{60}$ is ~10.34 Å, sufficient host–guest interactions are expected to be present between $C_{60}$ with **5** or **6**. However, the cavity of **7** is considered too large to allow it to form a stable host–guest complex with a $C_{60}$ molecule. It was synthesized and examined for the purpose of comparison.

**Single-molecule conductance.** The STMBJ technique was performed using a Au tip and a Au(111) single-crystal substrate to measure the single-molecule conductance of the metallocycles and their host–guest complexes with $C_{60}$ under ambient conditions (see the Methods section). The lone pair electrons of the free pyridine nitrogen atoms of metallocycles can bind to the Au tip and substrate to form molecular junctions by repeatedly moving the STM tip into and out of contact with the gold substrate functionalized with the supramolecular metallocycles[21,44]. Conductance–distance traces were recorded during the stretching process for each sample, and plateaus in these traces were related to the formation and breakdown of single-molecule junctions. Thousands of conductance traces were used to construct a conductance histogram, where the peak position measures the average conductance of a single molecule (Fig. 2).

Figure 2b plots the typical conductance vs. distance traces of the free metallocycles **5**, **6**, **7**, and those in the presence of $C_{60}$, denoted as $[5 + C_{60}]$, $[6 + C_{60}]$, and $[7 + C_{60}]$, respectively. A conductance plateau was typically observed for the free metallocycles and the conductance followed the order as **5**> **6**> **7** (3.1, 1.4, and $0.6 \times 10^{-5}$ $G_0$, respectively; see also in Supplementary Fig. 14 and Table 1). In contrast to the free metallocycles, the conductance–distance traces of $[5 + C_{60}]$ and $[6 + C_{60}]$ display two conductance plateaus at different conductance levels. The low-conductance plateaus center ~$10^{-5}$ $G_0$, where are about the same as that for the free metallocycles. However, the high-conductance plateaus observed for $[5 + C_{60}]$ are located at $2.8 \times 10^{-4}$ $G_0$ and $[6 + C_{60}]$ at $1.7 \times 10^{-4}$ $G_0$, respectively, suggesting the formation of the host–guest complex with $C_{60}$[28]. It is worth noting that different patterns of conductance plateaus were also observed for $[5 + C_{60}]$ and $[6 + C_{60}]$, including those with only the high- or low-conductance plateau and those showing discrete or smooth transition from the high-conductance plateaus to the low-conductance plateaus (Supplementary Fig. 15). In contrast, $[7 + C_{60}]$ exhibits only one conductance plateau at ~$0.61 \times 10^{-5}$ $G_0$, which is at the same level as the free **7** metallocycle. This suggests that no host–guest complex was formed between **7** and $C_{60}$ under the same measurement conditions. This conclusion is also supported by the DFT calculations.

As shown in Fig. 3, the conductance histograms of **5**, **6**, **7**, and $[7 + C_{60}]$ only reveal a single well-defined peak, while $[5 + C_{60}]$ and $[6 + C_{60}]$ exhibit two well-separated conductance peaks. The average conductance values and the yields of junction formation for the metallocycles and their host–guest complexes are summarized in Table 1. The control experiments with pure solvent or $C_{60}$ solution (10 μM) show no conductance peak in the

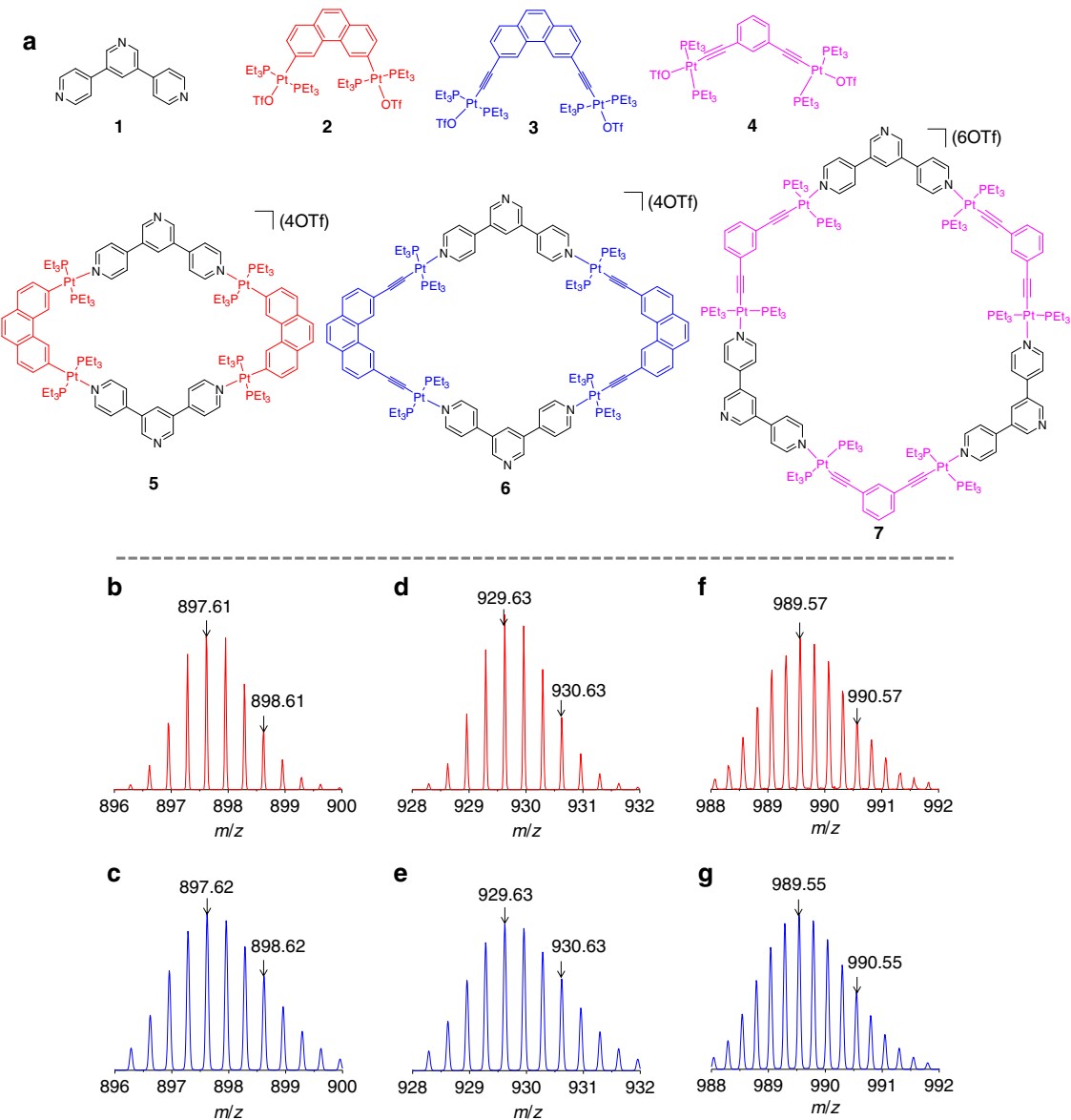

**Fig. 1** Compounds studied in this work and characterizations. **a** Molecular structures of **1**–**7**. **b**, **d**, **f** ESI-TOF-MS experimental data and **c**, **e**, **g** predicted isotope distribution of **b**, **c** [**5** − 3OTf]$^{3+}$, **d**, **e** [**6** − 3OTf]$^{3+}$, and **f**, **g** [**7** − 4OTf]$^{4+}$

histogram within the measured conductance range (Supplementary Fig. 20). It is worth noting that the concentration of $C_{60}$ in our experiments was low (10 μM), and after rinsing with copious amounts of solvent, the coverage of $C_{60}$ on Au surface was too low to be measured by break junction methods[45–50]. However, $C_{60}$ featured two conductance peaks at ~0.2 $G_0$ and 0.5 $G_0$, with a higher concentration of $C_{60}$ (7 mM), which is in agreement with reported values (Supplementary Fig. 19)[45,49].

These conductance trends suggest that the three metallocycles (**5**, **6**, and **7**), with different shapes and cavity sizes, possess different degrees of host–guest interactions with the $C_{60}$ guest. In contrast to [**7** + $C_{60}$], [**5** + $C_{60}$] and [**6** + $C_{60}$] show higher conductance values, which are greater than those of the free **5** and **6** molecules by about one order of magnitude. This provides direct evidence of large enhancement of charge transport when a $C_{60}$ guest is incorporated. This finding led us to explore in situ controlling of the host–guest interaction by mechanically stretching the molecular junction. Mechanical switching has been demonstrated in single molecules[35–38], but mechanical control of host–guest interactions is a new and attractive strategy. The 2D

conductance histograms and corresponding average plateau lengths of **5**, **6**, **7**, [**5** + $C_{60}$], [**6** + $C_{60}$], and [**7** + $C_{60}$] are shown in Fig. 4a–c and Supplementary Figs. 16, 18. The plateau lengths of conductance–distance traces represent the change in electrode separation, when a molecular junction is stretched from formation to breakdown, and is determined by the lifting and shape deformation of the metallocycle during the stretching process. The plateau lengths of the empty metallocycles (**5**, **6**) are comparable, but slightly larger than those of the host–guest complexes ([**5** + $C_{60}$] and [**6** + $C_{60}$]). Figure 4d–f shows the 1D histograms of metallocycle-$C_{60}$ complexes at different stretching distances. For [**5** + $C_{60}$], as the junction is stretched by a longer distance, the relative peak area of the lower conductance becomes increasingly larger, which is consistent with the transition from the high to low plateaus shown in individual traces in Fig. 2b. This observation suggests that the junction with **5** is still present after $C_{60}$ is released and the conductance switches back to the free host level. For [**6** + $C_{60}$], the relative peak area of the high conductance also decreases with stretching, but the distance dependence is substantially smaller than that for [**5** + $C_{60}$].

The distinct distance dependence gives a higher conductance band that persists for 0.2–0.3 nm and 0.5–0.6 nm for $[5 + C_{60}]$ and $[6 + C_{60}]$, respectively (Fig. 4d, e). This can be explained by the higher degree of flexibility for the relatively larger **6** than **5** to accommodate the $C_{60}$ guest. In contrast, $[7 + C_{60}]$ displays only one conduction peak at a different stretch distances, suggesting the absence of significant host–guest interaction.

"Pull and hold" experiments were also performed for $[5 + C_{60}]$ and $[6 + C_{60}]$ complexes. As demonstrated in Supplementary Fig. 17, the higher conductance states of $[5 + C_{60}]$ and $[6 + C_{60}]$ last for >0.1 s and 0.2 s, respectively. This differs from conventional break junction measurements of these complexes, where the high conductance states last for <0.03 s. These results suggest that the conductance switching from higher to lower level is mechanically dependent.

The maximum mechanical stretching force of the STM tip for the pyridine contact is ~0.8 nN[51], which is strong enough to induce the structural deformation of the metallocycles and thus the release of the $C_{60}$ guest. These results suggest that the driving force of forming these metallocycle-$C_{60}$ host–guest complexes is a weak noncovalent interaction, where the shape and cavity size of the metallocycle host play a vital role in the complexation. For metallocycle **5** with a cavity size (10.87 Å) comparable with the van der Waals diameter of $C_{60}$ (10.34 Å), a very subtle structural

deformation of the host by 0.1~0.2 nm could be destructive for the host–guest interaction. However, for metallocycle **6** with a moderately larger cavity size (14.10 Å), the host could stand a structural deformation as large as 0.5~0.6 nm, while still maintaining the host–guest complex. Such information can not be obtained on conventional studies of bulk materials.

To provide a more complete view of the overall break junction process over a wider conductance range, we measured the conductance from the quantum point contact formed between the electrodes to the noise level measured when the electrodes are widely separated using a logarithmic amplifier[52]. The resulting 1D and 2D histograms of **5**, **6**, $[5 + C_{60}]$, and $[6 + C_{60}]$ are shown in Supplementary Fig. 21. As summarized in Supplementary Table 1, the conductance values measured by logarithmic and linear amplifier are in agreement within 13.5%.

The distributions of relative displacement from the point of contact to junction breakdown are shown in Supplementary Fig. 22. After adding the Au–Au snap-back distance (0.5 nm), the junction breaking distances of **5**, $[5 + C_{60}]$, **6**, and $[6 + C_{60}]$ are 1.38 nm, 1.38 nm, 1.41 nm, and 1.42 nm, respectively, suggesting similar junction breakdown configurations for the metallocycles before and after encapsulation of $C_{60}$. From the DFT optimized molecular length of **5** (1.79 nm) and **6** (1.94 nm), the angle between the center axis of the metallocycle and the substrate plane is estimated to be ~50° for both metallocycles at junction breakdown. These results indicate that during the break junction measurements the metallocyclic molecules were lifted and distorted by the retreating STM tip.

**Spectroscopic studies.** UV−vis absorption spectral titration experiments were carried out to investigate the host–guest interactions in solution (Supplementary Figs. 23–25). For **5** and **7**, no distinct absorbance changes were observed upon adding the metallocycle into the $C_{60}$ solution. In contrast, the complexation of **6** with $C_{60}$ resulted in the appearance of a new shoulder band at 366 nm, which was consistent with a charge transfer (CT) absorption band[53]. The linear Benesi–Hildebrand plot suggests a 1:1 host–guest binding ratio of **6** with $C_{60}$ and the binding constant was calculated to be $4.1 \times 10^5 \, M^{-1}$. These results suggest that the driving force of forming the target host–guest complexes is derived from the donor–acceptor interactions between the electron-rich metallocylic skeleton and the electron-deficient of $C_{60}$[54]. The failure to observe potential CT absorption band from the complexation of **5** with $C_{60}$ is possibly caused by the relatively weak and shallow interaction between them. This is supported by the DFT calculation results discussed below. Although the absorption spectral analysis implies that the interaction between **5** and $C_{60}$ is weaker compared with that between **6** and $C_{60}$, the metallocycle **5** can still hold $C_{60}$ with a shallower host–guest configuration according to our DFT calculations. This allows the complexed molecular wire, either in the case of $[5 + C_{60}]$ or $[6 + C_{60}]$, to open up the $C_{60}$-related current path to enhance the conductance (see the section Transmission Calculations below).

**XPS characterizations.** In our previous work, Pt(II)-based self-assembled supramolecular rectangle, square, three-dimensional

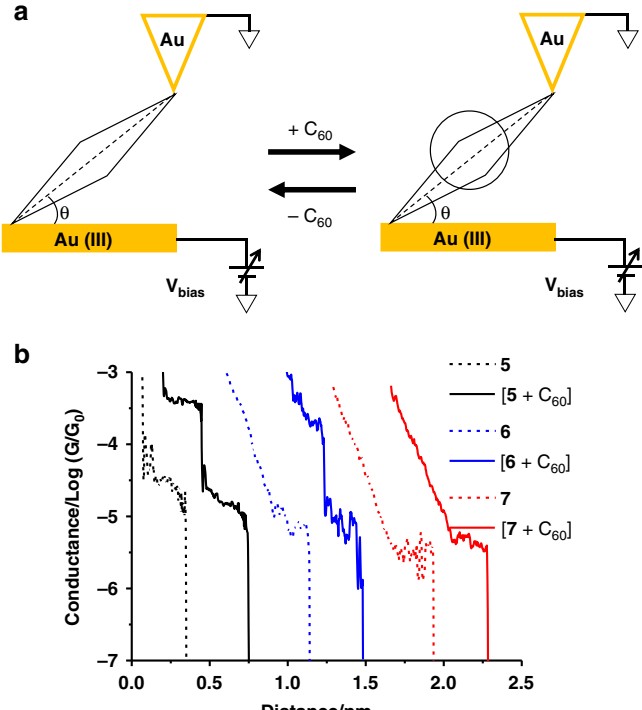

**Fig. 2** Conductance vs. distance traces for metallocycles and their host–guest complexes with $C_{60}$. **a** Schematic view of molecular junction (θ≈50°). **b** Representative conductance vs. distance traces of **5**, **6**, **7**, and $[5 + C_{60}]$, $[6 + C_{60}]$, $[7 + C_{60}]$

| Table 1 Conductance values and corresponding junction formation yield of complexes | | | | | | |
|---|---|---|---|---|---|---|
| **Complex** | **5** | $[5 + C_{60}]$ | **6** | $[6 + C_{60}]$ | **7** | $[7 + C_{60}]$ |
| **Conductance[a]** ($G_0$) | $(3.1 \pm 0.2) \times 10^{-5}$ | $(3.4 \pm 0.3) \times 10^{-5}$, $(2.8 \pm 0.1) \times 10^{-4}$ | $(1.4 \pm 0.1) \times 10^{-5}$ | $(8.9 \pm 0.3) \times 10^{-6}$, $(1.7 \pm 0.1) \times 10^{-4}$ | $(6.0 \pm 1.0) \times 10^{-6}$ | $(6.1 \pm 0.9) \times 10^{-6}$ |
| **Yield** | $(13 \pm 1)\%$ | $(12 \pm 2)\%$ | $(11 \pm 2)\%$ | $(12 \pm 1)\%$ | $(9 \pm 1)\%$ | $(10 \pm 0)\%$ |

[a]Error bars were calculated from three sets of experiments for each sample. $G_0 = 2e^2/h = 77.4 \, \mu S$ is the conductance quantum

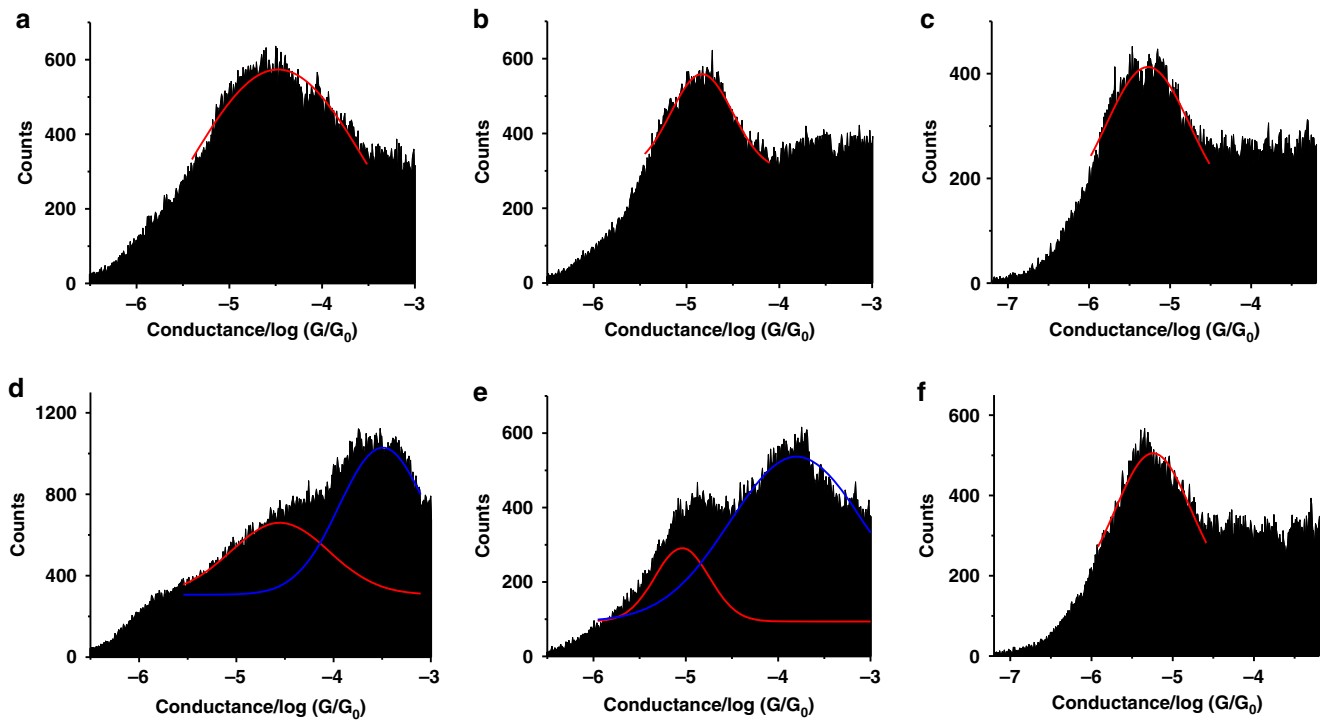

**Fig. 3** Conductance histograms of metallocycles and their host–guest complexes with $C_{60}$. Conductance histograms constructed from over 1000 individual traces: **a** for **5**, **b** for **6**, **c** for **7**, **d** for [**5** + $C_{60}$], **e** for [**6** + $C_{60}$], and **f** for [**7** + $C_{60}$]. The red and blue curves are Gaussian fits of the conductance peaks

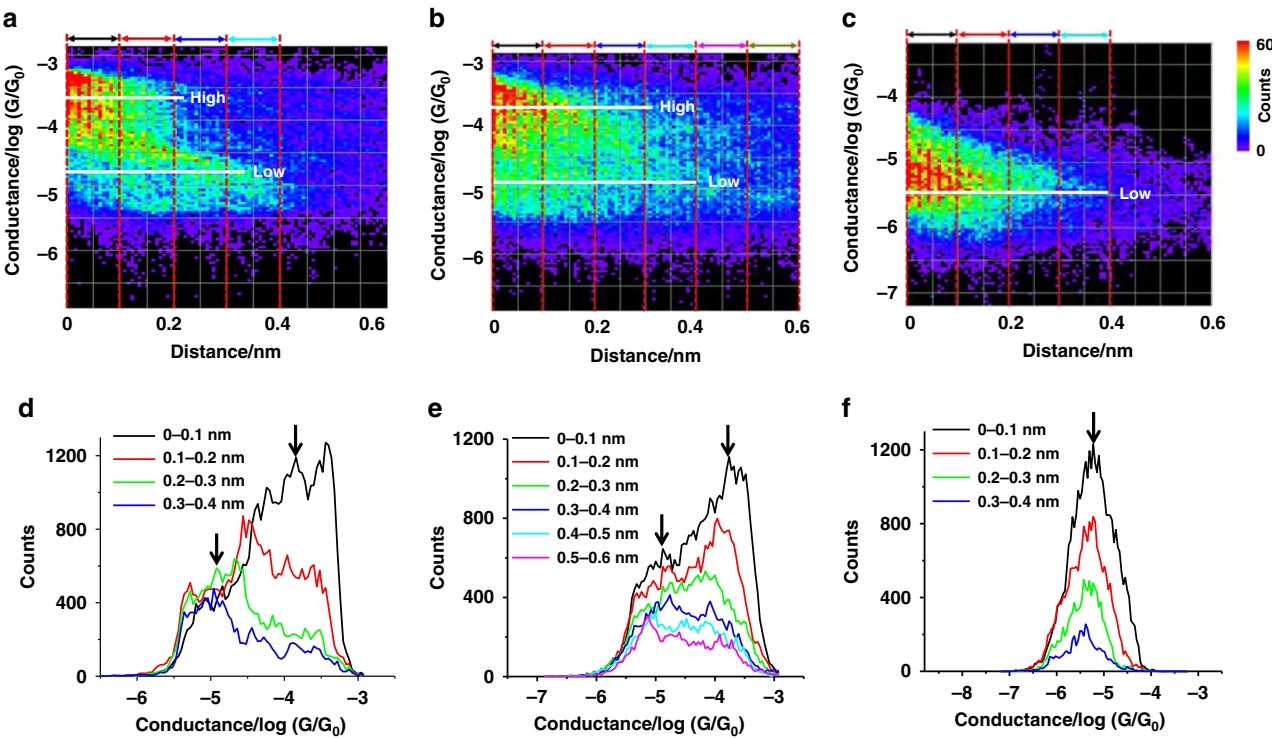

**Fig. 4** 2D conductance–distance histograms and 1D slices at different stretching distance of metallocycle-$C_{60}$ complexes. **a**, **b**, **c** 2D histograms. **d**, **e**, **f** 1D histograms. **a**, **d** for [**5** + $C_{60}$], **b**, **e** for [**6** + $C_{60}$], and **c**, **f** for [**7** + $C_{60}$]. Only the plateau region is included. The counts are represented by the color in 2D histograms. The white horizontal lines in 2D histograms and the black arrows in 1D slices indicate the higher and lower conductance levels

cage and prism have been investigated by scanning tunneling microscopy on Au(111) surfaces, and demonstrated to be intact after adsorption on Au(111) surfaces[55,56]. Herein, XPS characterizations were also carried out on monolayers on Au surfaces and corresponding powders of **5** and **6**. As shown in

Supplementary Figs. 26–29, the selected signals of Pt4f, P2p, N1s, and F1s are observed in both powders and monolayers. According to the further quantitative analysis (Supplementary Tables 2, 3), the atomic percentages of the elements of monolayers agree well with the powders compositions, suggesting that

the entire molecule remains absorbed on the Au surfaces. For example, the Pt/P/N/F atomic ratio of **5** monolayer (1.0/1.4/3.0/2.6) is consistent with that of the powder sample (1.0/2.1/2.7/2.7) within experimental error.

**AFM imaging**. AFM images of Au(111)/**5** and Au(111)/**6** are shown in Supplementary Fig. 30. The average heights of the films on Au(111) were determined with AFM[57], which are 1.2 nm and 1.3 nm for **5** and **6**, respectively, close to the STMBJ values.

**Theoretical calculations**. The host–guest complexes of **5** and **6** with $C_{60}$ were optimized by DFT calculations. The $C_{60}$ molecule is deeply embedded into the metallocycle framework of **6** (Supplementary Fig. 31). However, due to the smaller cavity size of **5**, the $C_{60}$ molecule only shallowly resides on one side of the metallocycle. This is in agreement with the above spectroscopic studies. The highest occupied molecular orbital (HOMO) and the lowest unoccupied molecular orbital (LUMO) plots of the [**5** + $C_{60}$] and [**6** + $C_{60}$] complexes are displayed in Fig. S32. The metallocycle skeleton mainly contributes to the HOMO and $C_{60}$ contributes to the LUMO, suggesting the presence of donor–acceptor interactions in the metallocycle-$C_{60}$ complexes.

In order to gain further insight into the conductance changes when the $C_{60}$ guest binds to the metallocycle host, the transmission spectra $T(E)$ were obtained by combining the DFT package SIESTA[58] and with the quantum transport code Gollum[59] (see details in the Methods section). Based on the relaxed geometries of the gold-molecule-gold junctions estimated from experimental nanogap distance in Supplementary Fig. 22, the transmission functions of host–guest complexes (red curve) are increased relative to the pristine metallocyclic hosts (blue curve) over a range of energies within the HOMO-LUMO gap, indicating that the higher conductance originates from the additional current path due to $C_{60}$ and the smaller angles (indicated by the two crossing red-dashed lines in Supplementary Fig. 42) between pi-orbital of the anchor pyridine (perpendicular to pyridine plane) and the Au–N bond formed by uncoordinated Au and N of this pyridine[60] (Fig. 5). More geometrical details could be found in Supplementary Fig. 42. After removing $C_{60}$ from the host–guest complex junctions, while freezing the metallocycle, the transmission functions of the distorted hosts **5′** and **6′** (shown by the yellow curves in Fig. 5g, h) are lower than those of [**5** + $C_{60}$] and [**6** + $C_{60}$]. This indicates the presence of an extra current path in the complexes due to the presence of $C_{60}$. Figure 5g shows that the transmission of **5′** with smaller angles (~45°) is close to that of the undistorted **5** with larger angles (~65°), while the large conductance discrepancy between red ([**5** + $C_{60}$]) and yellow (**5′**) curves in Supplementary Fig. 33g is observed, where the molecules are ideally fully extended and

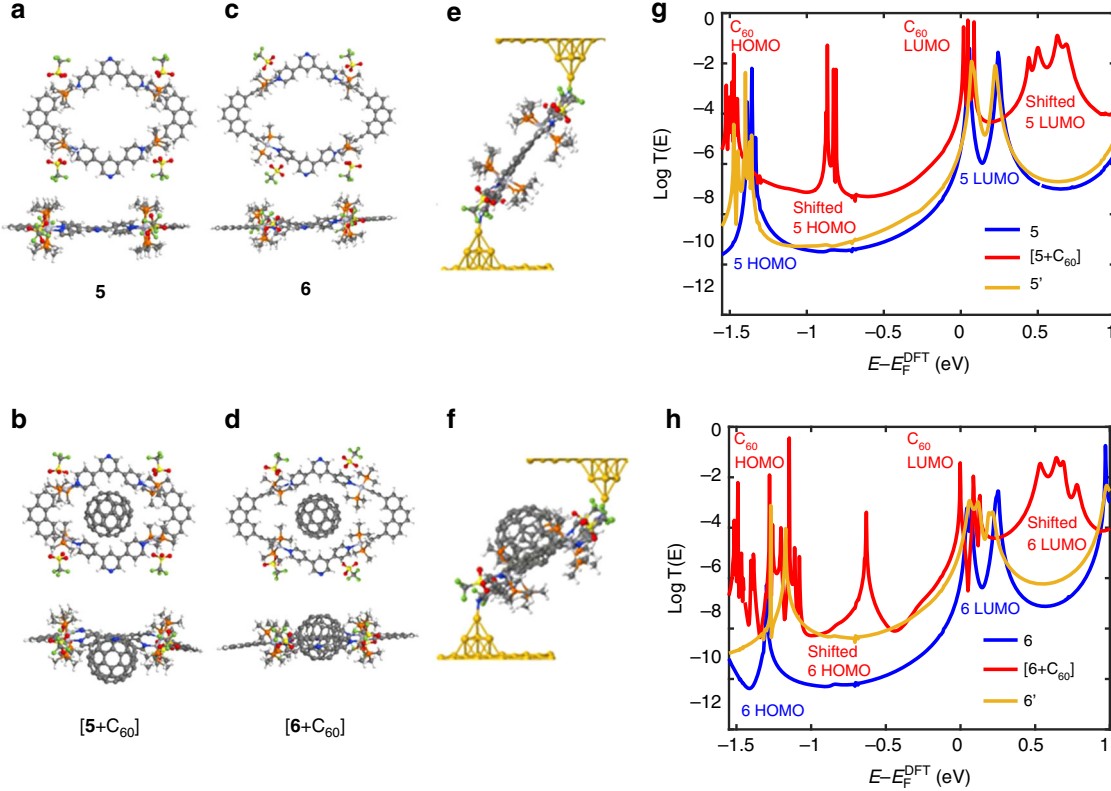

**Fig. 5** Gold/molecule/gold junctions quantum calculations. Top and lateral views of metallocycle cation with -OTF anions (CF3SO3−). **a** for **5**, **b** for [**5** + C60], **c** for **6**, **d** for [**6** + C60]. Single-molecule junctions containing CF3SO3− anions, where the terminal atom of the gold tip is bonded above and below to the nitrogen atom of the pyridine with a Au-N distance of 2.3 Å. **e** for **5**, **f** for [**5** + C60] where the angles between the Au-N bond formed by electrode-apex Au and N of the connected anchor pyridine and pi-orbital of this pyridine are around 45° and 65°, respectively after geometrical optimization. Quite similar junction geometries for **6** and [**6** + C60] are not shown which could be found in Supplementary Fig. 42. Gold atoms in the electrodes are colored yellow. In the molecule, gray, blue, and white represent carbon, nitrogen, and hydrogen atoms; green, light yellow, red, large grep, and orange represent fluorine, sulfur, oxygen, platinum, and phosphorus atoms. **g**, Transmission functions of **5**, [**5** + C60] and the distorted **5** (denoted as **5′**) which is obtained from the junction of [**5** + C60] by removing C60. **h**, Transmission functions of **6**, [**6** + C60] and the distorted **6** (denoted as **6′**) which is obtained from the junction of [**6** + C60] by removing C60. The resonances of red curve are labeled by red words while the blue curve correspond to blue words. Corresponding room temperature electric conductances are shown in Supplementary Fig. 42

nearly the same contact geometries are adopted. This reveals that the distortion of the host (shown in Fig. 5b) and the different contact angles caused by constraints can both have a significant effect on the conductance. On the other hand, Fig. 5h reveals that the transmission of **6′** with smaller tilt angles is higher than that of **6** with larger angles, while Supplementary Fig. 33h shows that the transmission of **6′** is much closer to that of **6**. This demonstrates that the smaller distortion of this host (shown in Fig. 5d) has a less significant effect on the conductance, but the angle difference has a great impact. Since the transmission of **5′** is lower than **5** in Supplementary Fig. 33g, while the transmission of **6′** is slightly higher than that of **6** in Supplementary Fig. 33h, we conclude that the distortion of the host can either increase or decrease the conductance and the effect of distortion is more pronounced in **5′** than in **6′**. The electric conductances shown in Supplementary Fig. 42 (obtained from transmission functions via formula (2) of the Methods section) depend on the precise value of the Fermi energy, which is expected to lie within HOMO-LUMO gap. Over the small range of Fermi energies (shaded gray in Supplementary Fig. 42) near DFT-predicted value, the conductances are $\sim 10^{-4}$ and $\sim 10^{-6}$–$\sim 10^{-5}$ for the complexes and metallocycles, respectively, which are consistent with experiment.

To understand the origin of the resonances in Fig. 5g, h, the local densities of states (LDOS) for molecular junctions were evaluated (Supplementary Figs. 43–44 for junctions with constraints which are consistent with experimental electrode separations and Supplementary Figs. 35–36 for more ideal fully extended molecules in junctions). These allow us to conclude that the transmission resonances (red curves) of the host–guest complexes around 0 eV are mainly associated with the frontier orbitals of the $C_{60}$, due to charge transfer between the host and guest. In contrast with oligothiophene–TCNE charge transfer complexes[61], the host–guest charge transfer does not result in Fano resonances, because Fano resonances occur when a pendant orbital weakly couples to a current-carrying backbone and sits orthogonal to the current path. In our host–guest complexes, the bound states on the $C_{60}$ form part of the current path and therefore produce Breit–Wigner resonances, rather than Fano resonances. However, as discussed charge transfer in oligothiophene–TCNE complexes[61] can cause energy levels on the donor or acceptor to move toward the Fermi level and therefore the resonance (Fano or otherwise) tends to appear near $E_F$. This effect is clearly present in the red curves of Fig. 5, where the resonances associated with the $C_{60}$ LUMO are pinned close to the DFT-predicted Fermi energy. In contrast, the resonances between −1.0 and −0.8 eV for [**5** + $C_{60}$] and between −1.0 and −0.5 eV for [**6** + $C_{60}$] are due to states located on the host. Therefore, these HOMO resonances are sensitive to the gating effect of the negatively charged $C_{60}$ guest (see charge transfer in Supplementary Tables 4, 5). Consequently, they move down in energy as the $C_{60}$ is systematically moved away from the host (see details shown by the ideally fully extended junctions in Supplementary Fig. 34).

It is worth noting that the transmission functions could be changed by modulating the contact geometry between the gold electrode and the molecule, as shown by Supplementary Figs. 37, 39, and 40. The binding energy calculation shows −1.08 eV for pyramidal tip Au–N, −0.6 eV for atop Au–N with flat gold electrode, −0.53 eV for Au–N bond out of pyridine plane, which are significantly greater than $k_B T$ at room temperature and therefore the junctions shown in Supplementary Figs. 37, 39, and 40 are stable (see details in Supplementary Fig. 41). These calculations show that whereas the increase in conductance due to host–guest binding is independent of the shape of the electrodes, the predicted conductance values are not.

Three organoplatinum(II) metallocycles with precisely controlled shape and cavity size were synthesized by coordination-driven self-assembly. These compounds, together with $C_{60}$, have been successfully used to probe the strength and flexibility of the size-dependent host–guest interaction at the single-molecular level by STMBJ measurements. The molecular conductance of the metallocycles with a suitable cavity size could be enhanced by one order of magnitude by the noncovalent host–guest complexation with $C_{60}$. Upon mechanically stretching, the $C_{60}$ guest molecule can be released from the metallocycle-$C_{60}$-based junctions, accompanied by the conductance switching back to the free host level. Metallocycles with fine-tuned cavity size are shown to possess a different degree of flexibility to accommodate the $C_{60}$ guest molecule. These results demonstrate the capability of the STMBJ technique to probe guest–host interactions at the molecular level and the potential of metallocycles as conductance-switchable supramolecular bridges.

## Methods

**Immobilization of the metallocyclic host molecules (5/6/7) and $C_{60}$ guest molecules on Au electrode.** The metallocycles and $C_{60}$ were immobilized on Au (111) single-crystal electrode by two separate steps. (1) The electrochemically cleaned Au (111) single crystal electrode was first annealed with hydrogen flame briefly, immersed in acetone (Alfa Aesar, 99.5%) containing 10 μM **5/6** or methanol (Fisher Scientific, 99.8%) containing 10 μM **7**, incubated for 2 h, then rinsed with acetone (for **5** and **6**) or methanol (for **7**) and dried with nitrogen gas. (2) After the STMBJ measurements were performed on the metallocycle modified Au electrode, the resulting electrode was immersed in 1,1,2,2-tetrachloroethane (Alfa Aesar, 98%) containing 10 μM $C_{60}$, incubated for 1 h, then rinsed with 1,1,2,2-tetrachloroethane and dried with nitrogen gas for further STMBJ measurements.

**Conductance measurement of single-molecule junctions.** The break junction measurements[21] were carried out by a scanning tunneling microscope (Nanoscope E, digital Instruments) and a STM scanner (Molecular Imaging). The molecules modified on Au (111) single-crystal electrode was measured in the air with a small bias (0.1 V) applied between the STM substrate (Au (111) single-crystal electrode) and the STM tip that was freshly prepared by cutting a gold wire (0.25 mm diameter, 99.5%). The STM tip was repeatedly brought into contact and retracted from the substrate, during which thousands of conductance distance traces were collected to construct conductance histograms. For each analyte, the STM conductance measurement was performed three time as repeating experiments.

**Theory.** DFT calculations were conducted using the B3LYP exchange correlation function and implemented in the Gaussian 09 package. The electronic structures were optimized using a general basis set with the Los Alamos effective core potential LANL2DZ basis set for Pt and 6–31 G* for other atoms[62]. No symmetry constraints were used in the optimization (nosymm keyword was used). Conductance calculations and corresponding geometry optimizations were performed by using the DFT code SIESTA[58], with a local density approximation (LDA functional), double-ζ polarized basis set for elements in molecules, double-ζ for gold element, cutoff energy of 200 Ry, 0.02 eV/A force tolerance, and 1 × 1 k points in the transverse directions. To compute the electrical conductance of the molecules, they were each placed between pyramidal gold electrodes. For each structure, the transmission coefficient $T(E)$ describing the propagation of electrons of energy E from the left to the right electrode was calculated by GOLLUM[59] code based on the resulting Hamiltonian and overlap matrices from SIESTA and the formulae of transport theory:

$$T(E) = Tr\big[\Gamma_L(E)G(E)\Gamma_R(E)G^{\dagger}(E)\big] \quad (1)$$

Where $\Gamma_{L,R}(E) = i(\Sigma_{L,R}(E) - \Sigma_{L,R}^{\dagger}(E))/2$ is the imaginary part of the self-energies $\Gamma_{L,R}(E)$. $\Gamma_{L,R}$ determines the width of transmission resonances and self-energies describe the contact between the molecule and left (L) and right (R) electrodes. G is the retarded Green's function of the molecule in the presence of the electrodes. The conductance is extracted from the transmission spectrum and evaluated by the following formulae:

$$G = G_0 \int_{-\infty}^{+\infty} dE\, T(E)\left(-\frac{\partial f(E)}{\partial E}\right) \quad (2)$$

where $G_0 = 2e^2/h$ is the conductance quantum; h is the Planck's constant; e is the charge of a proton; $f(E) = (1 + \exp(E - E_F/k_B T))^{-1}$ is the Fermi–Dirac probability distribution function, $E_F$ is the Fermi energy. Finally, the room temperature electrical conductance was computed from the formula (2).

## Data availability

The data that support the findings of this study are available from the article and Supplementary Information files, or from the corresponding authors upon request.

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

## Acknowledgements
N.T. thanks NSFC grant 21773117. C.L. thanks EPSRC grants EP/P027156/1, EP/N03337X/1, EP/N017188/1 and the ECH2020 FET Open project 767187 "QuIET" and the EU project Bac-to-Fuel. X.L. is thankful for financial support from NIH (R01GM128037). Y.-W.Z. thanks the National Natural Science Foundation of China (grants 21872154 and 21472196) and the Strategic Priority Research Program of the Chinese Academy of Sciences (grant XDB12010400) for funding support. P.J.S. thanks the U.S. National Institutes of Health (Grant R01 CA215157) for financial support. The authors would like to thank Prof. Wenjing Hong and State Key Laboratory of Physical Chemistry of Solid Surfaces at Xiamen University for accommodating the visit and providing the STM setup with logarithmic current-voltage converter for further experiments.

## Author contributions
J.-H.T., Y.S., J.Y.; Y.-W.Z. and P.J.S. conceived and designed the experiments. Y.L., Z.W.; H.W.[5] and N.T. performed single-molecule conductance measurements. Q.W., S.H. and C.L. contributed conductance calculations. K.T. contributed to XPS experiments. H.W.[7] and X.L. conducted the mass spectral measurements. C.L., X.W. and D.W. contributed to AFM measurements. All authors contributed to the analysis and interpretation of results and wrote the paper.

## Competing interests
The authors declare no competing interests.
