## [Peer Review File · Nature Communications]

Single-molecule level control of host-guest interactions in metallocycle-C₆₀ complexes

Tang et al.

Supplementary files

Supplementary Methods

General experimental details

All reagents were commercially available and used as supplied without further purification. ^1H NMR and ^{13}C NMR spectra were recorded in the designated solvents on a Varian Inova 500 MHz spectrometer. $^{31}\text{P}\{^1\text{H}\}$ NMR spectra were recorded on a Varian Unity 300 MHz spectrometer, using an external unlocked sample of 85% H_3PO_4 ($\delta = 0$) as reference. Mass spectra were recorded on a Micromass Quattro II triple-quadrupole mass spectrometer using electrospray ionization with a MassLynx operating system. Absorption spectra were recorded on a Hitachi U-4100 Spectrophotometer.

Synthesis of **5**

The mixture of **1** (2.33 mg, 10 μmol) and **2** (13.37 mg, 10 μmol) were added to mixed solvent of $\text{CH}_2\text{Cl}_2/\text{CH}_3\text{OH}$ (1 mL/4 mL). The system was stirred at 60 $^\circ\text{C}$ for 10 h. After cooling to room temperature, the solution was concentrated by N_2 flow and precipitated by adding diethyl ether. The pale-white solid product was obtained by centrifugation (14 mg, 95%). ^1H NMR (500 MHz, CD_2Cl_2): δ 9.34 (s, 4H), 9.17 (d, $J = 5.5$ Hz, 4H), 9.13 (s, 2H), 8.76 (d, $J = 5.5$ Hz, 4H), 8.71 (s, 4H), 8.65 (d, $J = 5.5$ Hz, 4H), 8.18 (d, $J = 6.0$ Hz, 4H), 7.62-7.65 (m, overlapped, 12H), 1.36–1.48 (m, overlapped, 48H), 1.14–1.26 (m, overlapped, 72H). $^{31}\text{P}\{^1\text{H}\}$ NMR (121.4 MHz, CD_2Cl_2 , 298 K): δ 14.54 ppm (s, ^{195}Pt satellites, $^1J_{\text{Pt-P}} = 3321.3$ Hz). ESI-TOF-MS calcd for $[\text{M} - 3\text{OTf}]^{3+}$ (m/z): 897.62. Found: 897.61.

Supplementary Figure 1. ^1H NMR spectra of **1**, **2** and **5** in CD_2Cl_2 .

Supplementary Figure 2. $^{31}\text{P}\{^1\text{H}\}$ NMR spectra of **2** and **5** in CD_2Cl_2 .

Supplementary Figure 3. ¹H-¹H COSY spectrum of **5** in CD₂Cl₂.

Supplementary Figure 4. Mass spectrum of **5**.

Synthesis of **6**

1 (2.33 mg, 10 μmol) and **3** (13.85 mg, 10 μmol) were dissolved in $\text{CH}_2\text{Cl}_2/\text{CH}_3\text{OH}$ (1 mL/4 mL). The mixture was stirred at 60 $^\circ\text{C}$ for 10 h. The system was then concentrated by N_2 flow and precipitated by adding diethyl ether. The pale-white solid was collected by centrifugation to give product **6** (15 mg, 98%). ^1H NMR (500 MHz, CD_2Cl_2): δ 9.77 (s, 4H), 9.40 (s, 4H), 8.81 (d, $J = 5.7$ Hz, 8H), 8.52 (d, $J = 5.9$ Hz, 10H), 7.82 (d, $J = 8.1$ Hz, 4H), 7.70 (s, 4H), 7.55 (d, $J = 8.0$ Hz, 4H), 1.91–1.94 (m, overlapped, 48H), 1.24–1.31 (m, overlapped, 72H). $^{31}\text{P}\{^1\text{H}\}$ NMR (121.4 MHz, CD_2Cl_2 , 298 K): δ 16.37 ppm (s, ^{195}Pt satellites, $^1J_{\text{Pt-P}} = 2852.2$ Hz). ESI-TOF-MS calcd for $[\text{M} - 3\text{OTf}]^{3+}$ (m/z): 929.62. Found: 929.63.

Supplementary Figure 5. ^1H NMR spectra of **1**, **3** and **6** in CD_2Cl_2 .

Supplementary Figure 6. $^{31}\text{P}\{^1\text{H}\}$ NMR spectra of **3** and **6** in CD_2Cl_2 .

Supplementary Figure 7. ^1H - ^1H COSY spectrum of **6** in CD_2Cl_2 .

Supplementary Figure 8. Mass spectrum of 6.

Synthesis of **7**

1 (2.33 mg, 10 μmol) and **4** (12.85 mg, 10 μmol) were dissolved in CH_3OH (5 mL). The mixture was stirred at 60 $^\circ\text{C}$ for 10 h. The system was then concentrated by N_2 flow and precipitated by adding diethyl ether. The pale-white solid was collected by centrifugation to give **7** (9.0 mg, 91%). ^1H NMR (500 MHz, CD_3OD): δ 9.24 (d, $J = 2.20$ Hz, 6H), 8.88 (t, $J = 4.70$ Hz, 14H), 8.21-8.23 (m, overlapped, 13H), 7.09-7.20 (m, overlapped, 12H), 1.87-1.93 (m, overlapped, 72H), 1.15-1.27 (m, overlapped, 108H). $^{31}\text{P}\{^1\text{H}\}$ NMR (121.4 MHz, CD_3OD , 298 K): δ 17.03 ppm (s, ^{195}Pt satellites, $^1J_{\text{Pt-P}} = 2860.26$ Hz). ESI-TOF-MS calcd for $[\text{M} - 4\text{OTf}]^{4+}$ (m/z): 989.55. Found: 989.57.

Supplementary Figure 9. ^1H NMR spectra of **1**, **4** and **7** in CD_3OD .

Supplementary Figure 10. $^{131}\text{P}\{^1\text{H}\}$ NMR spectra of **4** and **7** in CD_3OD .

Supplementary Figure 11. ^1H - ^1H COSY spectrum of **7** in CD_3OD .

Supplementary Figure 12. Mass spectrum of 7.

Supplementary Figure 13. CPK model of the DFT-optimized structures of **5** (a), **6** (b), **7** (c), C_{60} (d).

Supplementary Figure 14. Individual conductance versus distance traces measured for **5** (a), **6** (b), **7** (c) modified on Au substrate.

Supplementary Figure 15. Individual conductance versus distance traces measured for **[5 + C₆₀]** (a), **[6 + C₆₀]** (b), **[7 + C₆₀]** (c) modified on Au substrate.

Supplementary Figure 16. 2D conductance-distance histograms of free metallocycles **5** (a), **6** (b) and **7** (c). Only the plateau region is included. The counts are represented by the color in 2D histograms.

Supplementary Figure 17. ‘Pull and hold’ experiments of [5 + C₆₀] (a) and [6 + C₆₀] (b). The white horizontal lines indicate the higher and lower conductance levels. In the break junction measurement, the tip movement was halted for 1 s once a junction formation is detected.

Supplementary Figure 18. Histograms of the plateau length in conductance versus distance traces of 5 (a), [5 + C₆₀] (d), 6 (b), [6 + C₆₀] (e), 7 (c), 7 incubated with C₆₀ (f). Red curves are Gaussian fittings of the plateau length peaks. The numbers mark the peak position of Gaussian fittings.

Supplementary Figure 19. Conductance histogram of C_{60} , where the Au substrate was incubated with C_{60} (7 mM) in 1,1,2,2-tetrachloroethane for 2 hours, and transferred to pure 1,1,2,2-tetrachloroethane for 1 minute, then rinsed with DI water. The blue arrows mark the conductance peaks from single C_{60} molecules, and the red arrows marks the G_0 peak.

Supplementary Figure 20. STM break junction control measurements. The conductance histograms measured at different conductance range. (a, c, e), Au substrate incubated with acetone for 1 hour, rinsed, and dried with nitrogen. (b, d, f), the resulting substrate from (a, c, e) incubated in 1,1,2,2-tetrachloroethane solution of

10 μM C_{60} for one hour, rinsed and dried with nitrogen. No feature conductance peak except for the G_0 peak shows up in these histograms, indicating no analytical feature from the solvent of macrocycles, and no binding between the C_{60} and the Au electrode after rinsing.

Supplementary Figure 21. 1D and 2D conductance histograms of metallocycles and their host-guest complexes with C_{60} , measured with logarithm amplifier controlled STM. a, e for **5**, b, f for **6**, c, g for [**5** + C_{60}], d, h for [**6** + C_{60}]. The red and blue curves are Gaussian fits of the conductance peaks.

Supplementary Table 1. The conductance measured by logarithm and linear amplifiers.

Complex	Conductance/ Log(G/G_0) (Linear)	Conductance/ Log(G/G_0) (Log)	δ
5	-4.51	-4.42	2.00%
[5 + C_{60}]	-4.47	-4.77	6.71%
	-3.55	-3.25	8.45%
6	-4.85	-5.08	4.74%
[6 + C_{60}]	-5.05	-5.06	0.198%
	-3.77	-3.26	13.5%

Supplementary Figure 22. The relative displacement distributions from electrodes' point of contact to . (a) junction break of **5**, (b) the sharp drop of higher plateau for [**5** + C₆₀], (c) junction break of [**5**+C₆₀], (d) junction break of **6**, (e) the sharp drop of higher plateau for [**6**+C₆₀], (f) junction break of [**6** + C₆₀]. The red curves are Gaussian fits.

Supplementary Figure 23. Absorption spectra of metallacycle/C₆₀ = 1/1 ($c = 2 \times 10^{-6}$ M) in acetone measured after different stored time. **a** for [**5** + C₆₀], **b** for [**6** + C₆₀], **c** for [**7** + C₆₀].

Supplementary Figure 24. Absorption spectra of C_{60} (2×10^{-6} M) with incremental addition of **5** ($2 - 20 \times 10^{-7}$ M) in acetone. The data was measured after being stored at rt for 8 h.

Supplementary Figure 25. a, Absorption spectra of C_{60} (2×10^{-6} M) with incremental addition of **6** ($2 - 20 \times 10^{-7}$ M) in acetone. The data was measured after being stored at rt for 8 h. **b,** Corresponding Benesi-Hildebrand plot fitted at 366 nm, using the Benesi-Hildebrand equation $1/\Delta A = 1/\Delta A_{\text{sat}} + 1/(\Delta A_{\text{sat}} K [\text{host}])$, Where ΔA was the change in absorbance of C_{60} upon addition of **6** and ΔA_{sat} was the maximum absorbance difference. The binding constant (K) was evaluated graphically by plotting $1/\Delta A$ versus $1/[\mathbf{6}]$. The experimentally observed data were linearly fitted and the K values were obtained from the slope and intercept of the line.

Supplementary Table 2. XPS survey of **5**.

atomic %	5	
	on Au	powder
Pt4f	0.89	0.75
P2p	1.27	1.60
N1s	2.65	2.01
F1s	2.31	2.03
atomic ratio of (Pt/P/N/F)	1.0/1.4/3.0/2.6	1.0/2.1/2.7/2.7

Supplementary Table 3. XPS survey of **6**.

atomic %	6	
	on Au	powder
Pt4f	0.78	0.75
P2p	1.07	1.50
N1s	3.00	2.36
F1s	2.17	3.73
atomic ratio of (Pt/P/N/F)	1.0/1.4/3.8/2.8	1.0/2.0/3.1/5.0

Supplementary Figure 26. Pt4f, P2p, N1s and F1s XPS spectra of **5** on Au.

Supplementary Figure 27. Pt4f, P2p, N1s and F1s XPS spectra of the powder of **5**.

Supplementary Figure 28. Pt4f, P2p, N1s and F1s XPS spectra of **6** on Au.

Supplementary Figure 29. Pt4f, P2p, N1s and F1s XPS spectra of the powder of **6**.

Supplementary Figure 30. AFM images in both tapping and contact modes and corresponding height changes of Au(111)/**5** (a, b, c), Au(111)/**6** (d, e, f). AFM images of (b, e) were measured after the scratching of the modified surface by the contact mode.

Supplementary Figure 31. DFT-optimized structures of [5 + C₆₀] and [6 + C₆₀]. DFT calculated structures of [5 + C₆₀] **a** (top view), **b** (side view). DFT calculated structures of [6 + C₆₀] **c** (top view), **d** (side view).

Supplementary Figure 32. The HOMO and LUMO distributions of a, b for [5 + C₆₀] and c, d for [6 + C₆₀]. Hydrogen atoms are omitted for clarity.

Supplementary Figure 33. Gold/molecule/gold junctions quantum calculations. Top and lateral views of metallocycle cation with $-\text{OTF}$ anions (CF_3SO_3^-). a for **5**, b for $[\mathbf{5}+\text{C}_{60}]$, c for **6**, d for $[\mathbf{6}+\text{C}_{60}]$. Single-molecule junctions containing CF_3SO_3^- anions, where the terminal atom of the gold tip is bonded above and below to the nitrogen atom of the pyridine with a Au-N distance of 2.3 Å. e for **6**, f for $[\mathbf{6}+\text{C}_{60}]$. Gold atoms in the electrodes are coloured yellow. In the molecule: grey, blue, white represent carbon, nitrogen, and hydrogen atoms; green, light yellow, red, large grey, orange represent fluorine, sulphur, oxygen, platinum, phosphorus atoms. g, Transmission functions of **5**, $[\mathbf{5}+\text{C}_{60}]$ and the distorted **5** (denoted as **5'**) which is obtained from the junction of $[\mathbf{5}+\text{C}_{60}]$ by removing C_{60} . h, Transmission functions of **6**, $[\mathbf{6}+\text{C}_{60}]$ and the distorted **6** (denoted as **6'**) which is obtained from the junction of $[\mathbf{6}+\text{C}_{60}]$ by removing C_{60} .

Supplementary Figure 34. Configurations and transmission functions corresponding to four stages during the release of C_{60} . a, Configurations and b, transmission functions of a series of $[5+C_{60}]$ junctions. c, Configurations and d, transmission functions of a series of $[6+C_{60}]$ junctions. For the configurations shown in the red, purple, light purple and yellow rectangular boxes, the corresponding transmission curves are shown red, purple, light purple and yellow plots respectively. The red and yellow transmission curves are the same with those shown in Fig. S33.

energy window	-1.7 ~ -1.4eV		-0.1 ~ 0.3eV	
5				
$[5+C_{60}]$				energy window	-1.5 ~ -1.4eV C_{60} HOMO	-1.2 ~ -0.8eV Shifted 5 HOMO	0 ~ 0.1 eV C_{60} LUMO	0.3 ~ 0.6 eV Shifted 5 LUMO

Supplementary Figure 35. Local Density of States (LDOS) of the junctions within different energy windows for fully extended **5** and [**5**+C₆₀] in gold/molecule/gold junctions (Fig. S33e, f).

Energy window	-1.6 ~ -1.4eV		0 ~ 0.4eV	
6				
[6 +C ₆₀]				energy window	-1.5 ~ -1.2eV C ₆₀ HOMO	-1 ~ -0.5eV Shifted 6 HOMO	0 ~ 0.2eV C ₆₀ LUMO	0.5 ~ 0.8eV Shifted 6 LUMO

Supplementary Figure 36. Local Density of States (LDOS) of the junctions within different energy windows for fully extended **6** and [**6**+C₆₀] in gold/molecule/gold junctions (Fig. S33e, f).

Supplementary Table 4. Net charges of isolated complex **5**, [**5**+C₆₀], **6**, [**6**+C₆₀] and corresponding fully extended junctions (Fig. S33e, f).

	ISOLATED COMPLEXES		JUNCTIONS	
5	0		-0.39	
[5 +C ₆₀]	0		-0.77	
	5 : -0.23	C ₆₀ : 0.23	5 : -0.29	C ₆₀ : -0.48
6	0		-0.41	
[6 +C ₆₀]	0		-0.86	
	6 : -0.16	C ₆₀ : 0.16	6 : -0.24	C ₆₀ : -0.62

Supplementary Figure 37. Geometries of the junctions with different contact details and corresponding transport properties.

Supplementary Figure 38. Geometries of **6**, **[6+C₆₀]**, **6'** which corresponds to the transmission spectra in Fig. S33h.

Supplementary Figure 39. Configurations for molecule **5** and complex **[5+C₆₀]** and the corresponding transmission spectra. a. Transmission functions as the function of energy. b. Conductance curves as the function of Fermi energy referring to that estimated by DFT for **5** and **[5+C₆₀]**.

Supplementary Figure 40. Configurations for molecule **6** and complex **[6+C₆₀]** and the corresponding transmission spectra. a. Transmission functions as the function of energy. b.

Conductance curves as the function of Fermi energy referring to that estimated by DFT for **6** and [**6**+C₆₀].

Binding energies to gold electrodes.

The article [1] considers a 4-4' bipyridine molecule attached to gold electrodes terminated by an adatom, dimer, trimer, pentamer or a pyramid. It demonstrates that the binding energy depends on the angle between the bipyridine long axis and the bond between the pyridine nitrogens and under-coordinated gold atoms on the tip of the electrodes. Tilting the nitrogen–gold bond out of the pyridine plane results in a decreased junction binding energy, from 1.36 eV for a vertical junction to 0.70 eV when tilted to 30 degrees. The binding energy of pyridyl anchors to gold electrodes was also studied in article [2]. On the other hand, these studies ignore important dispersion forces. When these are included the binding energy to a flat gold (111) surface increases substantially to around 0.78 eV [3]. As examples of these possibilities, we have presented results for both pyramidal (Fig. S33 and Fig. S37 of SI) and flat electrodes (Fig. S39, S40 of SI) and find that in both cases, the predicted host-guest interactions are consistent with our experiments. We have also calculated the binding energies to each electrode of the two geometries below, using the LDA(CA) exchange-correlation functional and counterpoise method [4] which eliminates basis set superposition errors (BSSE). Binding energies are calculated using the formula $E_b = E_{AB} - E_{a,B} - E_{A,b}$. Even without the inclusion of dispersive forces, all binding energies are significantly greater than $k_B T$ at room temperature and therefore the junctions shown in Figs. S37, S39, S40 are stable.

Supplementary Figure 41. Binding energies for tip-flat geometry (a) and tip-tip tilted geometry (b).

Supplementary Figure 42. Schematics of gold/molecule/gold break junctions during the pulling process and corresponding transport properties. a. Initial structure corresponding to high conductance of [5+C₆₀]. b. Optimal structure after geometrical relaxation. c. Initial structure of **5** or [5+C₆₀] after C₆₀ falls out. d. Optimal structure after geometrical relaxation for structure c. e. Initial structure corresponding to high conductance of [6+C₆₀]. f. Optimal structure after geometrical relaxation. g. Initial structure of **6** or [6+C₆₀] after C₆₀ falls out. h. Optimal structure after geometrical relaxation for structure g. i.

Corresponding conductances as the function of fermi level relative to that given by DFT for **5**, [**5**+C₆₀], **5'** which is obtained from the optimal junction (shown in b) of [**5**+C₆₀] by removing C₆₀. j. Corresponding conductances as the function of fermi level relative to that given by DFT for **6**, [**6**+C₆₀], **6'** which is obtained from the optimal junction (shown in f) of [**6**+C₆₀] by removing C₆₀.

energy window	-1.6 ~ -1.3eV		0 ~ 0.4eV	
5				
[5 +C ₆₀]				energy window	-1.6 ~ -1.4eV C ₆₀ HOMO	-1. ~ -0.7eV Shifted 5 HOMO	0 ~ 0.3 eV C ₆₀ LUMO	0.3 ~ 1 eV Shifted 5 LUMO

Supplementary Figure 43. Local Density of States (LDOS) of the junctions within different energy windows for tilted **5** and [**5**+C₆₀] in gold/molecule/gold junctions (Fig. S42d, b).

Energy window	-1.4 ~ -1.0 eV		0 ~ 0.5eV	
6				

[6+C ₆₀]				energy window	-1.6 ~ -1.3eV C ₆₀ HOMO	-0.8 ~ -0.5eV Shifted 6 HOMO	0 ~ 0.2eV C ₆₀ LUMO	0.5 ~ 0.8eV Shifted 6 LUMO

Supplementary Figure 44. Local Density of States (LDOS) of the junctions within different energy windows for tilted **6** and [6+C₆₀] in gold/molecule/gold junctions (Fig. S42h, f).

Supplementary Table 5. Net charges of isolated complex **5**, [5+C₆₀], **6**, [6+C₆₀] and corresponding tilted junctions (Fig. S42d, b and h, f).

	ISOLATED COMPLEXES		JUNCTIONS	
5	0		-0.2	
[5+C ₆₀]	0		-0.74	
	5 : -0.23	C ₆₀ : 0.23	5 : -0.18	C ₆₀ : -0.56
6	0		-0.16	
[6+C ₆₀]	0		-0.80	
	6 : -0.16	C ₆₀ : 0.16	6 : -0.17	C ₆₀ : -0.63

Supplementary References:

[1] Su Ying Quek, Maria Kamenetska, Michael L. Steigerwald, Hyoung Joon Choi, Steven G. Louie, Mark S. Hybertsen, J. B. Neaton and Latha Venkataraman, Nat. Nanotech. 2009, 4, 230–234.

[2] Wenjing Hong, David Zsolt Manrique, Pavel Moreno-García, Murat Gulcur, Artem Mishchenko, Colin J. Lambert, Martin R. Bryce, and Thomas Wandlowski, J. Am. Chem. Soc. 2012, 134, 2292–2304.

[3] D. Mollenhauer, N. Gasto, E. Voloshina and B. Paulus, J. Phys. Chem. C 2013, 117, 4470-4479.

[4] S. F. Boys and F. Bernardi, *Mol. Phys.* 1970, 19, 553–566.